# A Review on Berry Seeds—A Special Emphasis on Their Chemical Content and Health-Promoting Properties

**DOI:** 10.3390/nu15061422

**Published:** 2023-03-15

**Authors:** Natalia Sławińska, Katarzyna Prochoń, Beata Olas

**Affiliations:** Department of General Biochemistry, Faculty of Biology and Environmental Protection, University of Lodz, 90-236 Lodz, Poland

**Keywords:** berry, health, phenolic compound, phytochemical, seed

## Abstract

Berries are important components of the human diet, valued for their high content of nutrients and active compounds. Berry seeds are also important objects of scientific investigation as, in some cases, they can have a higher concentration of certain phytochemicals than other parts of the fruit. Moreover, they are often byproducts of the food industry that can be reused to make oil, extracts, or flour. We have reviewed available literature related to the chemical content and biological activity of seeds from five different berry species—red raspberry (*Rubus idaeus* L. and *Rubus coreanus* Miq.), strawberry (*Fragaria x ananassa*), grape (*Vitis vinifera* L.), sea buckthorn (*Hippophae rhamnoides* L.), and cranberry (*Vaccinium macrocarpon* Ait.). We have searched various databases, including PubMed, Web of Knowledge, ScienceDirect, and Scopus. Last search was conducted on 16.01.2023. Various preparations from berry seeds are valuable sources of bioactive phytochemicals and could be used as functional foods or to make pharmaceuticals or cosmetics. Some products, like oil, flour, or extracts, are already available on the market. However, many preparations and compounds still lack appropriate evidence for their effectiveness in vivo, so their activity should first be assessed in animal studies and clinical trials.

## 1. Introduction

Fruits are important components of the human diet, serving as a source of nutrients and bioactive phytochemicals. Among them, berries are especially valued for their high content of nutrients and active compounds [1]. Some studies have shown that a diet rich in berries has a positive impact on health due to their high content of phytochemicals [2,3,4]. Berry seeds are also interesting targets for scientific investigation. Their chemical content and biological activity are still being explored by researchers from around the world. Post-harvest processing by the agricultural and food industries is also an important aspect that can affect the content and activity of berries and must be taken into consideration to find the best ways of introducing new seeds or seed products to the human diet [5].

Lately, the interest in byproducts generated during food processing is rising. Making use of materials that would otherwise be seen as waste and discarded allows for the maximization of available resources and can lead to the production of new foods with high nutritional values. At the same time, it reduces problems associated with waste disposal [6]. Berries can be processed in many different ways, producing byproducts such as skin, pulp, and seeds. Developing new ways to reuse left-over plant material would significantly expand the market for berry products, benefiting producers and consumers [7,8]. During berry juice production, the liquid is separated from the remaining purée, which contains the seeds and the peel. It is possible to separate the seeds from the puree and then use them to obtain oil, for example by cold pressing, which preserves the bioactive compounds contained within. These oils are rich in essential fatty acids (EFAs), antioxidants, carotenoids, and vitamin E. EFAs are essential fatty acids that are not synthesized by the human body, so supplying them with food is crucial. Some berry byproducts, like pomace from cranberries and grapes, are already utilized. They are used to make various extracts or oils that show multiple activities, for example antioxidant or antibacterial. Grape seeds have a particularly high content of EFAs, vitamins E, and polyphenols. Other berry seeds that are used to make oil include raspberries, blackberries, blueberries, and strawberries. They are used as food or components of cosmetics, for example, creams, shampoos, bath oils, lipsticks, and many other products [6,7,8,9,10].

Numerous plant extracts and phytochemicals are being investigated as potential agents that could help with the treatment and prevention of common diseases, like cancer, diabetes, or cardiovascular diseases. Increased consumption of phenolic compounds has been linked with a decreased occurrence of diseases related to oxidative stress. As most phenolic compounds show some degree of antioxidant or pro-oxidant activity, plants are a good source of compounds with potential beneficial effects that could be used as therapeutic agents. For example, many phenolics have shown anti-cancer properties thanks to either antioxidant (chemoprevention through reduction of DNA damage) or prooxidant (induction of cancer cell death through generation of reactive oxygen species) activity [11,12,13].

Recently, our review paper has shown not only phytochemical characteristics but also biological properties of selected seeds that can be consumed in two ways—eaten directly with whole fruit (for example, black raspberry and blackberry) or separately (including hemp and sacha inchi) [14]. The present work is a review of the available literature related to the chemical content and biological activity of seeds from five different berry species—red raspberry (*Rubus idaeus* L. and *Rubus coreanus* Miq.), strawberry (*Fragaria x ananassa*), grape (*Vitis vinifera* L.), sea buckthorn (*Hippophae rhamnoides* L.), and cranberry (*Vaccinium macrocarpon* Ait.). Out of these, red raspberry, strawberry, sea buckthorn, and strawberry fit the common definition of the word ‘berry’ (a soft, juicy fruit that is sweet, sour, or tart), while grape fits only the botanical definition [15]. Their biological properties are presented and discussed in the context of various biological activities, including anti-cancer, anti-hyperlipidemic, anti-obesity, and others. We have chosen to review the seeds of species whose whole berries or pulp have well-documented health-promoting properties. We have searched various databases, including PubMed, Web of Knowledge, ScienceDirect, and Scopus; extra papers were identified by manually reviewing the references. The search was restricted to English-language publications. The last search was conducted on 16.01.2023. The main search criteria included a combination of “berry”, “seed”, and “health”.

## 2. Characterization of Berries

Botany defines berries as fleshy, seed-containing fruits formed from a single ovary that do not contain a stone and whose pericarp is divided into three layers [1,15]. They vary in terms of chemical composition—the type and concentration of substances depend on the species and variety, environmental conditions during growth, ripeness at the time of harvest, storage, and further processing [5]. Examples of edible berries are blueberries, cranberries, and bilberries (genus *Vaccinium*), raspberries, blackberries, and cloudberries (genus *Rubus*), gooseberries, black and red currants (genus *Ribes*), strawberries (genus *Fragaria*), chokeberries (genus *Aronia*), elderberries (genus *Sambucus*), and sea buckthorn berries (genus *Hippophae*) [5,16]. Some fruits, like strawberries, raspberries, or blackberries, are commonly referred to as berries due to their fleshy structure, even though they are composed of many smaller fruits and do not fit the botanical definition [1]. Berries can be eaten as fresh fruit or in the form of processed products like juice, jam, and wine. They can also be made into dietary supplements, for example, in the form of extracts [5,16].

## 3. Chemical Content of Berries and Berry Seeds

Berries are a source of fiber and sugars (like glucose and fructose), as well as iron, calcium, manganese, magnesium, phosphorus, potassium, zinc, selenium, sodium, and copper. They contain vitamins C, A, and E, which are known for their antioxidant activity. Among berries, blackcurrants and sea buckthorn are known as especially good sources of vitamin C [17,18,19]. One of the most important berry polyphenols are anthocyanins, which are responsible for their blue, violet, or red color. The most popular anthocyanins found in berries are cyanidin, delphinidin, petunidin, peonidin, malvidin, and pelargonidin. The next group of phenolics present in berries are proanthocyanidins—a group of condensed flavan-3-ols, especially abundant in strawberries, blueberries, and chokeberries. Oil pressed from berry seeds can have high phenolic content as well—for example, grape seed oil contains large amounts of epicatechins and catechins [18,20,21]. Phenolic acids are also important components of berries. Chlorogenic acid, ellagic acid, and gallic acid can be encountered most often. Ellagic acid makes up approximately half of all phenols in cranberries and raspberries [16]. Flavanols and their glycosides, which usually contain glucose, galactose, rhamnose, arabinose, or rutinose, are also present. Stilbenes, like resveratrol, pterostilbene, and piceatannol, are important components of berries, too. Grapes and red currants are especially rich sources of resveratrol [18,22]. Berry seeds also contain phytosterols like campesterol, stigmasterol, and β-sitosterol, which have cardioprotective potential (Table 1 and Table 2) [7]. The main groups of chemical compounds, including the secondary metabolites in berry seeds, are presented in Figure 1. Chemical structures of selected compounds are shown in Figure 2.

**Table 1 nutrients-15-01422-t001:** Chemical content of berry seeds.

Chemical Content of Berry Seeds			
Type of Compound	Compound	Berries	Citation
Phytosterols	campesterol	raspberry	[7,23]
		strawberry	
		sea buckthorn	
		cranberry	
	stigmasterol	raspberry	[7]
		strawberry	
		cranberry	
	β-sitosterol	raspberry	[7,14]
		strawberry	
		sea buckthorn	
		cranberry	
	Δ^5^-avenasterol	raspberry	[7,23]
		strawberry	
		sea buckthorn	
	Δ^7^-avenasterol	raspberry	[7]
		strawberry	
		cranberry	
Tocopherols and tocotrienols	α-tocopherol	raspberry	[7,14,24]
		strawberry	
		cranberry	
		sea buckthorn	
	α-tocotrienol	cranberry	[24]
	β-tocopherol	raspberry	[24,25]
		cranberry	
	γ-tocopherol	raspberry	[7,24,25]
		strawberry	
		cranberry	
	γ-tocotrienol	raspberry	[7,24]
		cranberry	
	δ-tocopherol	raspberry	[7,24]
		strawberry	
	β-carotene	raspberry	[23,26]
		sea buckthorn	
	zeaxanthin	sea buckthorn	[27]
Anthocyanins	cyanidin	raspberry	[14]
		grape	
	peonidin	raspberry	[14]
Flavanols	epicatechin	raspberry	[28,29]
		grape	
	catechin	raspberry	[14]
		grape	
Flavonols	kaempferol	grape	[14]
		sea buckthorn	
Vitamins	vitamin C	sea buckthorn	[23,24]
	vitamin A	sea buckthorn	[23]
Phenolic acids	chlorogenic acid	grape	[14]
	ellagic acid	raspberry	[30]
		strawberry	
	gallic acid	raspberry	[14,31]
		sea buckthorn	
Stilbenes	resveratrol	raspberry	[14]
		grape	

**Table 2 nutrients-15-01422-t002:** The content of sterols and fatty acids in cold-pressed berry seed oils [7,21,32,33].

Chemical Content	Oil from Berry Seeds				
	Cranberry	Raspberry	Strawberry	Sea Buckthorn	Grape
Fatty acids (g/100 g)					
Lauric acid	0.08	-	-	-	0.01
Myristic acid	0.08	0.07	0.05	0.14	0.05
Palmitic acid	5.38	2.43	4.32	9.27	6.6
Stearic acid	1.25	0.90	1.68	2.47	3.5
Oleic acid	25.30	10.87	14.55	25.02	14.3
Linoleic acid	37.68	53.67	42.22	37.33	74.7
Linolenic acid	30.09	31.68	36.48	23.13	0.15
Arachidonic acid	0.07	0.37	0.71	0.44	0.40
Saturated fatty acids	6.88	3.82	6.85	13.39	10.4
Monounsaturated fatty acids	25.14	11.02	14.71	26.15	14.8
Polyunsaturated fatty acids	67.98	85.16	78.44	60.46	74.9
Sterols (%)					
Campesterol	4.2	4.5	5.4	4.0	9.3
Stigmasterol	1.3	1.2	2.3	17.0	10.8
β-sitosterol	60.4	79.6	71.1	66.5	67.4

**Figure 1 nutrients-15-01422-f001:**
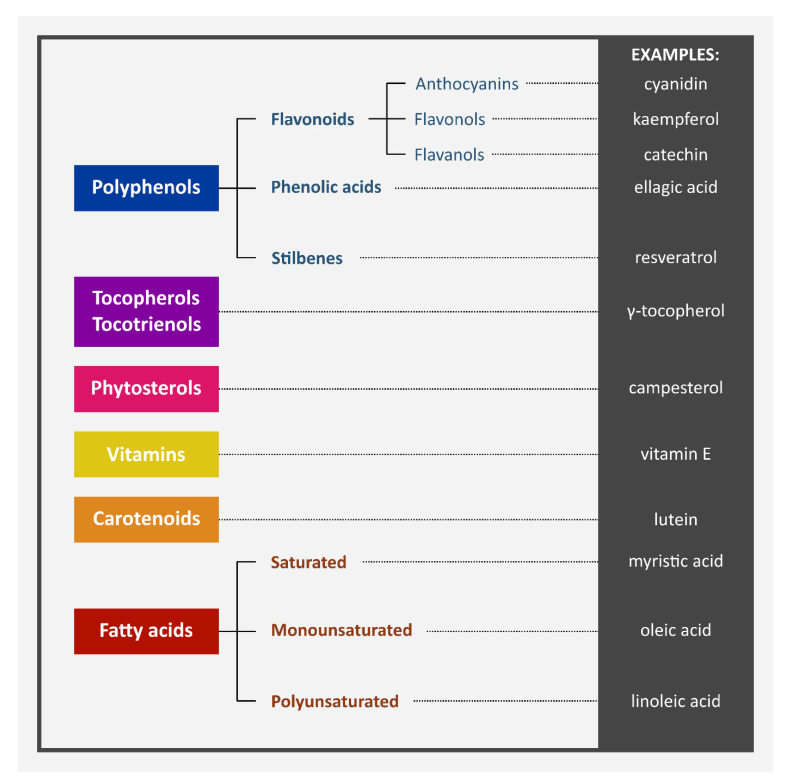
Main groups of chemical compounds (including secondary metabolites) in berry seeds. Compilation of data: [7,17,18,19,22,24,27,34].

**Figure 2 nutrients-15-01422-f002:**
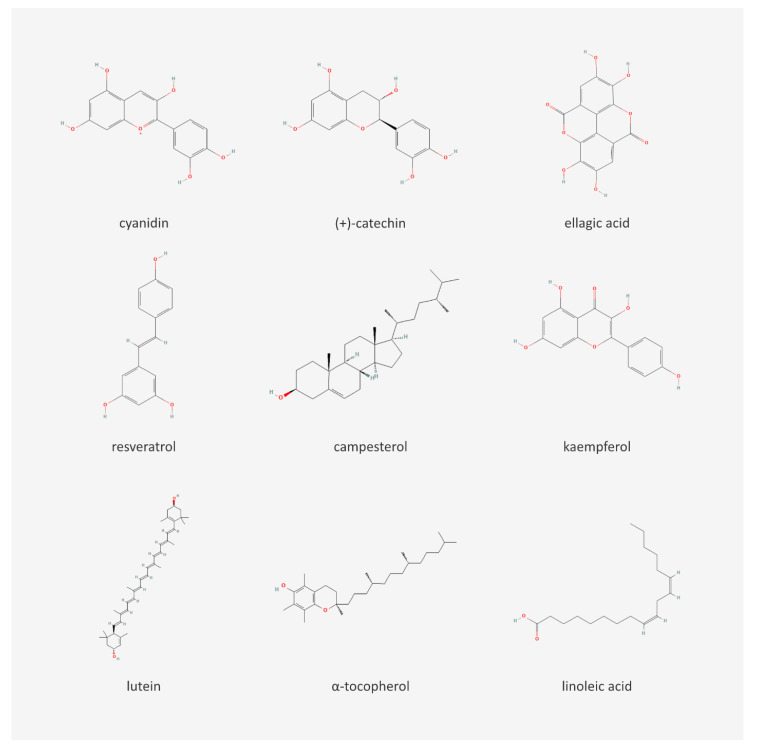
Chemical structures of selected compounds present in berry seeds [35].

## 4. Health-Promoting Properties of Berry Seeds—In Vitro and In Vivo Studies

### 4.1. Raspberry (R. idaeus L. and R. coreanus Miq.) Seeds

#### 4.1.1. In Vitro Studies

A study by Juranic et al. [36] has shown that water extracts from the seeds of five different raspberry cultivars (K81-6, Latham, Meeker, Tulameen, and Willamette) have antiproliferative potential against LS174 (human colon carcinoma) cells in vitro. The extract from the Willamette cultivar had the strongest effect. This antiproliferative activity has been linked to the presence of ellagic acid. At the same concentration, seed extracts had stronger activity than ellagic acid, which could be attributed to the synergistic effect of their other components. Red and black raspberries contained approximately 1500 μg/g dry weight (DW) of ellagic acid, more than cranberries (120 μg/g DW) or strawberries (630 μg/g DW). Moreover, more ellagic acid was present in the seeds (about 87.5%) than in the pulp (about 12.2%). Despite this, extracts from the fruits had a stronger antiproliferative effect than the seed extracts, though in the Willamette cultivar the differences were not that pronounced (the concentrations of the extract that resulted in 50% cell survival were 29.7% and 27.1%, respectively) [6,36,37]. Cho et al. also observed that black raspberry seeds are a source of ellagitannins, which have anticancer properties. They demonstrated this activity by arresting the cell cycle and inducing apoptosis in HT-29 colon cancer cells [38].

Raspberry seed oil showed antiproliferative activity as well. It inhibited the growth of the A-549 (human lung cancer) cell line [39]. In addition, results from Teng et al. indicate that raspberry seed oil can increase the expression of antioxidant enzymes (catalase (CAT), superoxide dismutase (SOD), and glutathione peroxidase (GPx) in HepG2 cells by suppressing extracellular signal-regulated kinase (ERK) and c-Jun *N*-terminal kinase (c-JNK) phosphorylation [40].

Recently, Grazjer et al. have studied two raspberry oils (0.5–10%): cold-pressed and extracted by supercritical CO_2_. They have observed that both tested oils are good candidates for non-toxic support in the therapy of selected human cancers, including colon adenocarcinoma (LoVo), breast cancer (MCF7), and lung cancer (A549) cell lines [41].

Choi et al. noted that raspberry seed extract (cyanidin-3-*O*-rutinoside was the major anthocyanin) has antioxidant potential. It inhibited the production of reactive oxygen species and reduced protein and DNA damage caused by hydroxyl radicals. Moreover, it inhibited lipid peroxidation in a dose-dependent manner [42].

Other results indicate that cyanidin-3-glucoside and gallic acid have inhibitory effects against viruses. The authors suggest that cyanidin-3-glucoside binds to murine norovirus-1 RNA polymerase. In addition, the results showed that gallic acid has antiviral properties against influenza A and B [43,44].

#### 4.1.2. In Vivo Studies

Kosmala et al. have observed that the addition of raspberry seeds to the diet improves the atherogenic index, especially due to the reduced concentration of triacylglycerol in rat serum [30].

Other studies showed that inclusion of ellagic acid (from raspberry seed flour) in the diet may normalize metabolic insults triggered by chronic intake of high sucrose in male C57BL/6 mice. This supplementation (0.03%, 12 weeks) attenuated the activation of endoplasmic reticulum stress and oxidative stress in the liver and decreased adipocyte inflammation [45].

Majewski et al. have studied the effect of ground raspberry seeds (whose main component was dietary fiber (64%) and the main phenolic compounds were ellagitannins (1.2%) and flavan-3-ols (0.45%)) on lipid profile, endothelium-intact vasodilation, and the enzymatic antioxidant status of plasma in normotensive and spontaneously hypertensive rats (n = 6). Tested animals were supplemented with 7% ground raspberry seeds for six weeks. Authors have observed that this supplementation neither modifies the body weight gain nor changes the daily feed intake of tested rats. Moreover, the typical lipid profile, including total cholesterol, total triglycerides, and high-density lipoprotein cholesterol, was also not changed. On the other hand, in normotensive rats, supplementation with the seeds modulated vascular functioning and increased vasodilation in response to acetylcholine [46].

Hendawy et al. studied the effect of cold-pressed raspberry seed oil on non-alcoholic fatty liver disease (NAFLD) in male Wistar rats [47]. NAFLD is a pathologic condition of the liver characterized by excessive deposition of fat in hepatocytes. It is correlated with type II diabetes, obesity, insulin resistance, elevated serum lipoproteins, and hypertension, and can lead to permanent liver damage. As NAFLD is a problem that affects 7–37% of adults and 20–30% of children worldwide, and currently used drugs (anti-obesity, antidiabetic, antioxidant, and antihyperlipidemic) can cause adverse effects, researchers are still searching for new ways of prevention and treatment of this disease [47,48]. Raspberry oil has a good ratio of *n*-6 and *n*-3 fatty acids (2–3:1) and contains γ-tocopherol, linolenic acid, polyphenols, and carotenoids, so it was chosen as a potential candidate [6,47,49]. Rats were fed 0.4 or 0.8 mL of raspberry seed oil (3 times a week for 8 weeks) in addition to a high-fat diet (HFD). Oil supplementation significantly inhibited the activity of alanine aminotransferase (ALT) and aspartate aminotransferase (AST). Blood glucose levels and insulin resistance decreased. Lipid peroxidation in the liver has been brought to a normal level, and the activity of SOD has increased. Moreover, liver inflammation and fat droplet accumulation induced by HFD were significantly decreased, especially by the dose of 0.8 mL. The researchers have also observed a significant reduction in the levels of nuclear factor kappa B (NF-κB) and leptin, as well as increased levels of peroxisome proliferator-activated receptor gamma (PPARγ) and adiponectin. Additionally, molecular docking showed that the identified fatty acids and tocopherols from raspberry seed oil could interact with PPARγ [47].

In another study, Fotschki et al. assessed the effect of raspberry seed oil on liver and intestine function, plasma lipid profile, oxidative stress, and inflammation in rats fed with a normal diet or a high-fat/low-fiber diet (HF/LF). Seven percent of the feed was replaced with raspberry seed oil. Oil supplementation increased β-glucosidase activity in the colonic digesta of rats fed with both types of diet. It improved the antioxidant status and fat proportion in the livers of rats fed with a normal diet and ameliorated the HF/LF diet-induced increase in plasma ALT and tumor necrosis factor alpha (TNF-α). The level of triglycerides decreased, but non-HDL cholesterol levels were unaffected. The effects of the oil on bacterial metabolism in the large intestine were inconclusive. Nevertheless, the study showed that raspberry seed oil can be a good source of essential fatty acids in the diet [49].

Lee et al. [50] studied the effect of α-linolenic acid-rich black raspberry seed oil on the metabolism of lipids in high-fat diet–induced obese and db/db mice. It reduced the levels of non-esterified fatty acids, triacylglycerol, and total cholesterol in the liver and serum of both the obese and db/db mice. The authors suggest that the mechanism of action of the tested oil includes inhibiting lipogenesis and promoting fatty acid oxidation.

### 4.2. Strawberry (F. x ananassa) Seeds

Strawberries (*F. x ananassa)* are well-known and popular fruits, valued for their high content of bioactive substances and health benefits. Not only fresh but also processed strawberries are good sources of polyphenols. The highest concentration (approximately 1429 mg/100 g DW) was found in the seeds. The main phenolic acid of strawberries is ellagic acid, and its content in the seeds is ten times higher than in the pulp [51]. Tiliroside is another important polyphenol present in strawberry seeds. It showed anti-inflammatory, hepatoprotective, anti-hyperglycemic, and antidiabetic activities. It also increased the expression of PPARα, which is correlated with its anti-obesity effect [52]. Seeds are especially valuable parts of strawberries; despite the fact that they comprise only 1% of the weight of the whole fruit, they contain up to 11% of its phenolics. Strawberries are consumed worldwide, and their seeds are eaten with the whole fruit. Grzelak-Błaszczyk et al. showed that defatted strawberry seeds are rich in protein, dietary fiber, polyphenols, vitamins, and minerals but have low sodium content. The composition of fruits obtained in three consecutive harvest seasons was similar [51].

#### 4.2.1. In Vitro Studies

Strawberry seed extract stimulated ceramide synthesis in stratum corneum in a 3-dimensional human epidermal equivalents (3DHEEs) culture model. Ceramides are the main lipids in the stratum corneum, which is the uppermost layer of the epidermis. They play a crucial role in water retention and skin barrier sustention. Treatment with 1 and 3 μg/mL of the extract significantly increased the total content of almost all ceramide species. Tiliroside had a similar effect, but it affected only two types of ceramides, and its activity was weaker overall [52].

#### 4.2.2. In Vivo Studies

Strawberry seed oil lowered the levels of oxidative stress in vivo. Administration of 0.8 mL of oil daily (for 5 weeks) decreased the activity of SOD and GPx in the blood of rats. Such changes suggest that after supplementation with strawberry oil, oxidative burden diminished and the lowered activity of antioxidant enzymes was sufficient to maintain balance [53].

Dietary strawberry seed oil (rich in α-linolenic acid and linoleic acid: 29.3% and 47.2% of total fatty acids, respectively) supplemented for 8 weeks affected metabolite formation in the distal intestine and normalized lipid metabolism in rats fed an obesogenic diet. This experiment included 32 male Wistar rats [54].

### 4.3. Grape (V. vinifera L.) Seeds

Grape (*V. vinifera* L.) seeds comprise 38% to 52% of the dry weight of the pomace left after fruit processing. They contain approximately 40% fiber, 16% oil, and 11% protein, as well as sugars and minerals. They are a valuable source of fatty acids and phenolics, which include catechin, epicatechin, gallic acid, and proanthocyanidins. Extracts and oils from grape seeds are consumed as dietary supplements or foods and are components of many cosmetic products [55].

#### 4.3.1. In Vitro Studies

The effect of grape seed extract (GSE) on human colorectal cancer cell lines (LoVo and HT29) was studied by Kaur et al. After 48 h of incubation, growth inhibition of human colorectal cancer cells was noted [56].

GSE had antibacterial activity as well. The minimal bactericidal concentration (MIC) against *Porphyromonas gingivalis* (ATCC 33277) was 4000 μg/mL, and against *Fusobacterium nucleatum* (ATCC 10953), 2000 μg/mL. Moreover, the extract inhibited biofilm formation and had antioxidant activity. Both *P. gingivalis* and *F. nucleatum* are involved in periodontitis and other acute periodontal disorders [57].

Grapeseed proanthocyanidins also showed anti-cancer properties. They inhibited the growth of the MCF-7 (breast cancer) cell line (*IC*_50_ = 0.10 μM). They induced cell cycle arrest at the G2/M phase and increased the number of apoptotic cells. Moreover, proanthocyanidins decreased the expression of epidermal growth factor receptor (EGFR), which is involved in cancer proliferation, migration, invasion, and angiogenesis [58]. In another study by Sun et al. grape seed proanthocyanidins inhibited the invasion of SCC13 (head and neck cutaneous squamous cell carcinoma) cells induced by epithelial growth factor (EGF), a known ligand and stimulator of EGFR [59].

#### 4.3.2. In Vivo Studies

Kong et al. studied the effect of grape seed proanthocyanidin extract (GSPE) on ischemia-reperfusion brain injury in mice. As oxidative stress plays an important role in ischemic stroke and proanthocyanidins can regulate this process, the authors hypothesized that GSPE might protect from ischemia-reperfusion brain damage. A dose of 50 mg/kg improved neurological function and decreased brain infarction. The number of mature, immunoreactive NeuN neurons was higher in mice treated with GSPE. At the same time, the expression of proapoptotic Bax and Bcl-2 was reduced. Moreover, other studies have shown that the antioxidant activity of GSPE is approximately fifty times higher than that of vitamins C and E. Grape seed proanthocyanidins extract showed promising neuroprotective activity, but the exact mechanism responsible for this effect is unclear and needs to be studied thoroughly [60].

Delimont and Carlson examined in vitro and in vivo studies that assessed the efficiency of grape seed extract in tooth decay prevention. The analysis showed that GSE has antibacterial activity and is particularly effective against *S. mutans*, which is responsible for the risk of caries. GSE could inhibit biofilm formation and colonization of *S. mutans* in the oral cavity. Studies also show that GSE supports tooth remineralization and increases resistance to enamel degradation. Despite promising research results, further studies of the effects of GSE on oral health are needed to develop the best methods of administering it to patients. Potentially, it could be used in mouthwashes and toothpaste in the future [61].

Kaur et al. studied the effect of 200 mg/kg of grape seed extract on athymic mice with a human colon carcinoma (HT29) tumor xenograft. GSE decreased proliferation and increased the rate of apoptosis from 18 ± 2% to 28 ± 2% [56]. Results from Li et al. demonstrated that lipophilic grape seed proanthocyanins have anticancer activity in HeLa cells. These compounds induced apoptosis through the intrinsic mitochondrial/caspase—mediated pathway [62].

Recently, results from Song et al. have demonstrated that procyanidins isolated from grape seeds may attenuate chemotherapy-induced cognitive impairment by decreasing matrix metalloproteinase-9 (MMP-9) activity. In this experiment, wild-type mice were supplemented with procyanidin (10, 20, or 40 mg/kg, p/o) 1 h before cisplatin treatment [63].

Abu Hafsa et al. [64] have noted that grape seeds (50 g/kg diet, for 98 days) decrease the oxidative stress induced by lindane-stimulated hepatotoxicity in rabbits (n = 24). For example, grape seed treatment increases the activity of catalase. Other results (in vitro and in vivo models) showed that grape seed proanthocyanidins (30 mg/mL or 400 mg/kg weight for 4 weeks) have radioprotective properties. These effects were associated with their antioxidant activity and inhibition of MAP kinase-activated protein kinase (MAPK) signaling pathways.

A systematic review and meta-analysis of randomized controlled trials by Anjom-Shoae et al. showed that there is a significant reduction in levels of low-density lipoprotein cholesterol (LDL-C) and total cholesterol in studies with <10 weeks of intervention and <300 mg/day of grape seeds [65]. Moreover, a review by Grohmann et al. indicates that grape seed supplementation (for 4 weeks) reduces systolic and diastolic blood pressure in subjects with hypertension or metabolic syndrome [66]. However, data on the effect of grape seeds on lipid profiles is still inconclusive.

More details about the different pharmacological properties of grape seeds and their potential health benefits were described by various authors, including Gupta et al., Liu et al. and others. For example, a review paper by Liu et al. suggests potential mechanisms of grape seed proanthocyanidins, emphasizing obesity prevention and treatment [67,68]. In a meta-analysis, Foshati et al. [69] showed that supplementation of grape seeds can inhibit lipid peroxidation. In addition, a systematic review by Coelho et al. [70] demonstrated that grape seeds have a positive effect on early glycation products. Asbaghi et al. [71] also observed similar effects of grape seed supplementation. Recently, Mahdipour et al. [72] have described the potential benefits of grape seeds in various neurological disorders. Their protective and therapeutic mechanisms of action might involve maintaining cellular proteostasis and antioxidant and anti-inflammatory activity. However, Bell et al. [73] suggest that younger, healthier people are less sensitive to grape seed doses < 400 mg relative to older or cognitively compromised populations.

Health-promoting properties (especially anti-inflammatory, anti-apoptotic, and antioxidant) that were observed not only in grape seeds but also in grape skin have been described in a review paper by Sochorova et al. [74].

### 4.4. Sea Buckthorn (H. rhamnoides L.) Seeds

Sea buckthorn (*H. rhamnoides* L.) is a shrub that grows on dry soils. Sea buckthorn seeds, fruits, and leaves are a source of valuable nutrients, such as polyunsaturated fatty acids, flavonoids, phenolic acids, catechins, proanthocyanidins, carotenoids, ascorbic acid, vitamins E, P, A and B, phytosterols, and tocopherols. These compounds could be utilized in the treatment of skin diseases and gastric ulcers and reduce the risk of cardiovascular diseases, cancer, and diabetes [27,34]. The oil pressed from sea buckthorn seeds has a high content of oleic acid (over 17%). It also contains palmitic acid, palmitoleic acid, stearic acid, linoleic acid, and linolenic acid [75].

#### 4.4.1. In Vitro Studies

The lipids and proteins of the skin are especially vulnerable to damage induced by ultraviolet (UV) radiation. For this reason, natural compounds with no adverse effects that can protect from UV-induced damage are sought after. Gęgotek et al. studied the effect of sea buckthorn seed oil on the correlation between redox balance, lipid metabolism, and the endocannabinoid system in UV-irradiated cells from different skin layers. UV radiation promotes the production of reactive oxygen species (ROS) and reduces the antioxidant capacity of skin cells, which results in a redox imbalance that leads to oxidative stress. Oxidative stress accelerates the degradation of collagen and the transformation of elastin, causing premature skin aging. Furthermore, UV-induced oxidative stress affects phospholipid metabolism, which in turn affects the endocannabinoid system, crucial in maintaining homeostasis. Sea buckthorn seed oil partially inhibited ROS formation in human fibroblasts and keratinocytes exposed to UVA and UVB radiation. The levels of non-enzymatic antioxidants, such as glutathione, thioredoxin, and vitamin A, were increased. Moreover, the researchers have found that the activity of nuclear factor erythroid 2-related factor 2 (Nrf2) transcription factor was increased, which upregulated the activity of antioxidant enzymes (GPx, glutathione reductase (GSSG-R), SOD, and thioredoxin reductase (TrxR)). As a result, the amount of lipid peroxidation products decreased and the level of endocannabinoid receptors increased. In conclusion, oil from sea buckthorn seeds significantly stimulated the antioxidant response of keratinocytes and fibroblasts and prevented UV radiation-induced damage, which makes it a potential candidate for new skin-protection products [76]. Moreover, Dudau et al. showed that sea buckthorn seed oil causes proliferation of both normal and dysplastic keratinocytes, under normal conditions and under the influence of UV radiation [77].

#### 4.4.2. In Vivo Studies

Sea buckthorn seed oil has shown anti-atherosclerotic properties. The study was conducted on healthy albino rabbits and rabbits fed a high-cholesterol diet for two months, which resulted in hypercholesterolemia. Administration of 1 mL of the oil to healthy animals for 18 days decreased LDL-C levels and the atherogenic index while increasing the concentration of high-density lipoprotein cholesterol (HDL-C), but total cholesterol and triglycerides remained unchanged. In rabbits fed a high-cholesterol diet supplemented with 1 mL of oil for 30 days, it lowered the level of LDL-C and increased the level of HDL-C in comparison to the control group, which was fed only a high-cholesterol diet with no oil. It has been suggested that these cardioprotective properties result from the presence of essential fatty acids, phytosterols, and vitamins A and E, which can act synergistically to improve cardiovascular system parameters [78].

Hao et al. studied the effect of sea buckthorn seed oil on blood cholesterol levels and gut microbiota in hamsters with hypercholesterolemia. The animals were divided into four groups. The first was a cholesterol-free control group; the second was a high-cholesterol control group (0.1% cholesterol in feed); and the third and fourth groups were fed a high-cholesterol diet with sea buckthorn seed oil replacing 50% or 100% of lard. The use of diets three and four lowered total plasma cholesterol by 20–22%, which was due to the reduction of cholesterol absorption and an increase in short-chain fatty acid production caused by changes in the intestinal microbiota [79].

A controlled trial by Vashishtha et al. has shown that sea buckthorn oil has positive effects on the cardiovascular systems of healthy and hypertensive/hypercholesterolemic humans. An oral dose of 0.75 mL for 30 days reduced systolic and diastolic blood pressure, serum total cholesterol, triglycerides, and LDL-cholesterol compared to baseline values [80].

In a study by Wang et al. sea buckthorn proanthocyanidins showed retinoprotective activity in rabbits subjected to visible light exposure. At the doses of 50 and 100 mg/kg/day, proanthocyanidins improved visual function, preserved retinal thickness, and decreased the levels of oxidative stress, inflammation, and apoptosis [81].

### 4.5. Cranberry (V. macrocarpon Ait.) Seeds

Cranberries (*V. macrocarpon* Ait.) are well-liked fruits, consumed for their taste and nutritional value. In the past, cranberry seed oil was used by sailors to prevent scurvy. It is used in the treatment of urinary tract infections, wounds, diarrhea, and diabetes, and has been shown to prevent tooth decay and stomach ulcers. Cranberries contain numerous active compounds. Their seeds can be pressed for oil, which is gaining popularity due to its well-balanced content of fatty acids and phytochemicals. It has a high concentration of flavonoids, tocopherols, tocotrienols, anthocyanins, and tannins. Cranberry seed oil has a much higher content of γ-tocotrienol, α-tocopherol, and γ-tocopherol than oils from the seeds of several different berry species, like blueberry, strawberry, or blackberry. Cranberries and their oil contain active compounds with antioxidant, anti-inflammatory, and anticancer effects, attributed to their phenolic acids, flavonoids, and proanthocyanidins. Moreover, they contain resveratrol, omega-3 fatty acids, and tocotrienols that have positive impacts on the cardiovascular system [82].

#### 4.5.1. In Vitro Studies

Cold-pressed cranberry seed oil showed good radical-scavenging activity against 2,2-azinobis-(3-ethylbenzthiazoline-6-sulfonic acid (ABTS) (22.5 ± 1.22 μmol Trolox equivalents (TE)/g) and 2,2-diphenyl-1-picrylhydrazyl (DPPH) radicals. The DPPH-scavenging capacity of cranberry oil was higher than that of carrot, black caraway, and hemp oils. 11.3 mg oil equivalent/mL depleted 95% of DPPH in the reaction mixture, which was comparable with the activities of vitamin C and α-tocopherol. Moreover, cranberry seed oil protected LDL from Cu^2+^-induced oxidation [83].

Cranberry flour had antioxidant activity as well. The oxygen radical absorbance capacity (ORAC) value was 110.5 ± 22.0 μmol TE/g flour. Moreover, flour extract showed anticancer properties. It inhibited the growth of HT-29 cells at both tested concentrations (3 and 6 mg/mL) [84].

#### 4.5.2. In Vivo Studies

The anti-hypercholesterolemic effect of the extract from cranberry seeds was tested on human volunteers with total blood cholesterol levels exceeding 200 mg/dL. In the fourth week of the study, the average cholesterol value for the entire experimental group decreased by 5.7 mg/dL [85].

More details about the various biological properties of berry seed preparations (in vitro and in vivo models) that were discussed in Section 4 are presented in Table 3. Moreover, food products from berry seeds and their health-promoting properties are shown in Figure 3. In addition, selected signaling pathways involved in the activity of berry seeds are presented in Figure 4.

**Table 3 nutrients-15-01422-t003:** Biological activity of berry seeds—in vitro and in vivo studies.

Berry	Preparation	Dose	Length of Study/ Incubation Time	Experimental Model	Biological Properties	References
In vitro studies						
Strawberry	Extract	0.1 and 0.3 µg/mL	4. 5. and 7 days	Human stratum corneum cells	Increased expression of PPARα	[52]
Cranberry	Flour	110.5 ± 22.0 μmol TE)/g flour	-	ORAC assay	Antioxidant	[84]
		3 and 6 mg/mL	4 days	HT-29 (colon cancer) cells	Antitumor	
	Oil	22.5 ± 1.22 μmol TE)/g	-	ABTS assay	Antioxidant	[83]
		11.3 mg oil equivalent/mL	-	DPPH assay		
		TBARS reduced by 2.8 mg/g oil	1 h	LDL oxidation assay		
Raspberry	Oil	30 µg/mL	48 h	A-549 (human lung cancer) cells	Inhibition of cell growth	[39]
	Extracts	*IC*_50_ = 27.1–49% of the extract (100 mg/mL)	72 h	LS174 (human colon carcinoma)	Antiproliferative	[36]
	Ellagitannins	5–30 μg/mL	24 and 48 h	HT-29 cells	Antitumor	[38]
	Fresh seed extract and wine seed extract	50–1000 μg/mL	1 h	HepG2 cell line	Antioxidant	[42]
		2.5–50 μg/mL	-	ORAC assay		
	Extract	10–1000 μg/mL	1–24 h	RAW 264.7 (mouse leukemic monocyte macrophage) and CRFK (Crandell Reese feline kidney) cells	Antiviral activity against FCV-F9 and MNV-1	[44]
	Extract	0.5–50 μg/mL	1 h	MDCK (Madin-Darby canine kidney) cells	Antiviral activity against influenza A and B	[43]
	Oil	0.5–2 mg/mL	1 h	HepG2 cells	Antioxidant	[40]
	Oil	0.5–10%	24–48 h	NHDF, LoVo, LoVo/DX, MCF7, MCF7/DX, A549 cells	Antioxidant, prooxidant, antitumor	[41]
Grape	Extract (GSE)	50 and 100 µg/mL	48 h	LoVo and HT29 (human colorectal cancer) cells	Inihibition of cell growth	[56]
	Extract (GSE)	2000 µg/mL	64 h	*P. gingivalis* ATCC 33277 and *F. nucleatum* ATCC 10953	Antibacterial	[9]
	Proanthocyanidins	10–80 μg/mL	1 h pre-incubation	TM3 (Mouse testicular stromal) and HIEC (Human small intestinal crypt epithelial) cells irradiated with 5 or 8 Gy	Antioxidant, radioprotective	[86]
	Proanthocyanidins	25–200 μg/mL	24 and 48 h	HeLa cells	Antitumor	[62]
	Proanthocyanidins	0.05–0.2 μM	12–48 h	MCF-7 cells	Antitumor (EGFR inhibition, antiproliferative)	[58]
	Proanthocyanidins	10–40 μg/mL	12–48 h	SCC13 cells	Antitumor (EGFR inhibition, invasion inhibition)	[59]
In vivo studies						
Sea buckthorn	Proanthocyanidins	50 and 100 mg/kg/day	14 days	Rabbits	Retinoprotective activity	[81]
	Oil	1 mL/day	18 days or 30 days	Healthy rabbits and rabbits with hypercholesterolemia	Anti-atherogenic activity	[78]
	Oil	50 and 100 g/day	42 days	Hamsters with hypercholesterolemia	Deceased total plasma cholesterol	[79]
	Oil	0.75 mL/day	30 days	Healthy and hypertensive/hypercholesterolemic humans	Decreased blood pressure, serum triglycerides, and cholesterol	[80]
Cranberry	Extract	15 mL/day	28 days	Humans with cholesterol level over 200 mg/dL	Decreased blood cholesterol	[85]
Strawberry	Oil	7% of diet	8 weeks	Rats fed with obesogenic diet	Anti-hyperlipidemic, negative changes to microbial metabolism in the distal intestine	[54]
Grape	Extract	200 mg/kg/day	56 days	Athymic mice with human colon carcinoma tumor xenograft	Anticancer activity	[56]
	Procyanidin extract (GSPE)	50 mg/kg/day	14 days	Mice with ischemia-reperfusion brain injury	Neuroprotective activity	[60]
	Procyanidin	1–40 mg/kg, administered 1 h before cisplatin treatment	30 days	Mice treated with cisplatin	Attenuation of chemotherapy-induced cognitive impairment	[63]
	Seeds	50 g/kg diet	98 days	Rabbits treated with lindane	Antioxidant	[64]
	Proanthocyanidins	400 mg/kg 1 h before radiation exposure, then fed continuously by drinking for 4 weeks (2 mg/mL)	56 days	Mice exposed to radiation (5, 8, or 27 Gy)	Radioprotective, antioxidant	[86]
Raspberry	Oil	7% of diet	56 days	Rats fed with HF/LF diet	Improved lipid metabolism	[49]
	Oil	0.4 and 0.8 mL/3 times a week	56 days	Rats fed with HFD	Potential anti-NAFLD effect	[47]
	Defatted seeds	6% of diet	4 weeks	Rats	Hypoglycemic, hypolipidemic, lower atherogenic index	[30]
	Flour	Equivalent of 0.03% ellagic acid	12 weeks	Mice fed with HFD or HFD with sucrose	Anti-hyperlipidemic, hypoglycemic, antioxidant, anti-inflammatory, normalization of adipocyte size	[45]
	Ground seeds	7 g/100 g of diet	6 weeks	Hypertensive and normotensive rats	Improved vascular function	[46]

## 5. Herb-Drug Interaction (HDI)

Herb-drug interaction (HDI) is an important aspect to consider when studying the effects of plant-derived compounds on the organism. Both herbal medicines and drugs can modify the activity of enzymes responsible for phase I and phase II reactions that are crucial in the metabolism of xenobiotics. Unfortunately, the absorption, distribution, metabolism, pharmacokinetics, efficiency, and safety of plant-derived medicinal products is often not studied enough, as they do not have to conform to the same regulations as synthetic drugs. As patients with chronic diseases who are often prescribed multiple drugs are more likely to use herbal medicines, studying their HDI, pharmacokinetics, and safety profile is of great importance [91].

For example, cranberry juice has been shown to inhibit the activity of cytochrome P4503A (CYP3A), which is involved in the metabolism of midazolam. Moreover, it decreased its absorption in the small intestine, which was studied in an in vitro model on Caco-2 cells. This was further confirmed in a clinical study, where cranberry juice lowered the absorption rate of midazolam and inhibited its intestinal metabolism [92].

Ortiz-Flores et al. have observed that (−)-epicatechin targets the pregnane X receptor (PXR), which is involved in xenobiotic metabolism. It induced translocation of PXR into the nucleus, which resulted in increased expression of Cyp3a11 in C2C12 cells [93]. On the other hand, the results of a clinical trial by Goey et al. showed that grape seed extract does not affect the pharmacokinetics of dextromethorphan in vivo, despite its inhibitory activity on CYP2D6 in vitro. This suggests that GSE could be safely combined with drugs that are metabolized by this enzyme [94].

## 6. Medicines and Commercial Products from Berry Seeds

Due to their unique chemical composition, berry seeds have become a common ingredient in cosmetics, food products, and dietary supplements. For example, raspberry seed oil is valued in the cosmetics industry. It is used in moisturizing creams as it contains high concentrations of vitamins A and E. Sometimes, it is added to makeup products, providing hydration, sun protection, and vitamins. It is also used in a variety of anti-wrinkle and skin-smoothing products [95].

Grape seed extract is recognized as safe by the US Food and Drug Administration (US FDA) and is used as a dietary supplement due to its strong antioxidant, anti-inflammatory, antimicrobial, anticancer, neuroprotective, anti-hyperlipidemic, antihypertensive, and anti-aging effects [87,88]. Apart from the extracts, grape seeds can be used to make powder (GSP) or flour, which are available on the market. Elkatry et al. have found that the addition of 10% GSP to balady bread increased its antioxidant capacity. At the same time, sensory characteristics remained acceptable [89]. Moreover, grape seed flour supplementation reduced the negative effects of HFD in mice. It modulated gut microbiota, reduced weight gain, and decreased plasma lipid levels [90]. Seed residues left after oil extraction can be used to obtain dietary fiber. The polysaccharides contained in grape seeds are known for their structure and physicochemical properties, such as antioxidant activity and water retention capacity. They are commonly referred to as antioxidant fibers [55,87].

Romanini et al. report that grape seed powder could be used as an alternative to bentonite—a type of clay used to prevent hazing during the wine-making process. Bentonite prevents hazing by removing pathogenesis-related proteins but can impair the quality, taste, and aroma of wine. GSP has been shown to reduce haze formation and could be used to minimize the use of bentonite [96].

The interest in developing new sea buckthorn products has been growing in recent years. Its berries, seeds, and leaves contain many biologically active substances that have the potential to be used for medicinal purposes, or to be utilized in the food and cosmetic industries [75,97]. Sea buckthorn is used for the production of juice, jam, confectionery, oil, dietary supplements, herbal teas, and cosmetics [34]. Sea buckthorn seed oil penetrates the epidermis easily and can improve blood circulation and skin oxygenation. It also protects against infections, decreases inflammation, prevents allergies, and slows down the aging process, which makes it a valuable addition to health and cosmetic products [27]. Moreover, sea buckthorn seed oil can help regenerate various skin conditions, such as eczema, burns, and wounds. It protects the skin from the damage induced by sun exposure, radiation therapy, and cosmetic laser therapy [98].

When used in cosmetics, the oil improves skin hydration and protects against transdermal water loss due to its content of fatty acids. Fatty acids also facilitate the transport of other oil components through the skin, allowing them to reach different layers of the epidermis. Its radical-scavenging activity protects the skin from oxidative damage, slowing the aging process. For this reason, it is often used in anti-wrinkle and anti-aging cosmetics. When applied to the skin, sea buckthorn seed oil soothes irritation, roughness, flaking, and itchiness. It can also be used to treat damage induced by exposure to UV radiation due to the presence of carotenoids and tocopherols. It is commonly used in peelings, masks, and depilation products because of its skin-smoothing effect. It also strengthens the hair, restores their elasticity, and prevents hair loss [99].

## 7. Conclusions

Berry seeds contain many bioactive compounds, including flavonoids, phenolic acids, stilbenes, phytosterols, essential fatty acids, and vitamins. They display a wide array of biological activities, for example, antioxidant, anti-inflammatory, antibacterial, anti-hyperlipidemic, neuroprotective, anti-tumor, or retinoprotective. As seeds are often byproducts of the food industry, reusing them in food additives, supplements, cosmetics, and pharmaceuticals is a good way to utilize available resources, reduce the amount of waste, and bring additional benefits to producers and consumers. In some cases, seeds are already used to make commercially available extracts and oils that are suitable for supplementation or added to cosmetics. There are multiple clinical trials (both completed and ongoing) that study the effect of grape seed extract on cardiovascular diseases, metabolic syndrome, or neurological disorders, but there are very few studies evaluating the efficiency of cranberry, raspberry, strawberry, and sea buckthorn seeds in vivo. There is a need for more clinical trials that would check the effectiveness and safety of less-known seeds, which would allow for full utilization of their potential.

## Figures and Tables

**Figure 3 nutrients-15-01422-f003:**
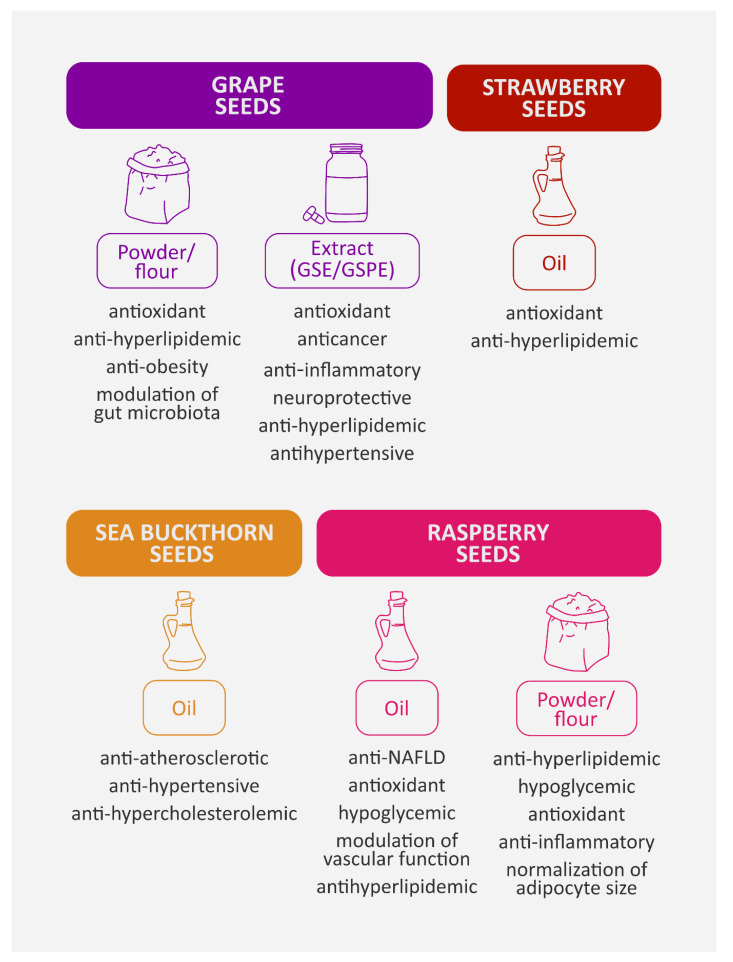
Selected food products from berry seeds and their biological activity. GSE—grape seed extract; GSPE—grape seed proanthocyanidin extract; NAFLD—non-alcoholic fatty liver disease. Compilation of data: [45,47,53,54,56,60,78,79,87,88,89,90].

**Figure 4 nutrients-15-01422-f004:**
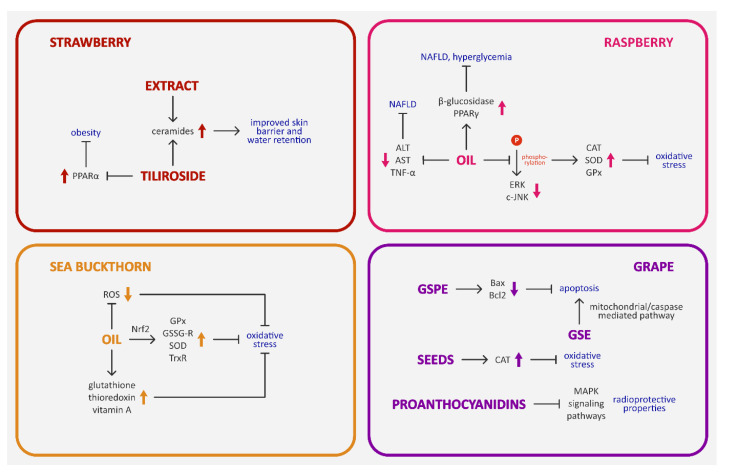
Selected signaling pathways involved in the activity of berry seeds and their health-promoting potential. ALT—alanine aminotransferase; AST—aspartate aminotransferase; Bax—apoptosis regulator Bax; Bcl-2—apoptosis regulator Bcl-2; CAT—catalase; c-JNK—c-Jun *N*-terminal kinase; ERK—extracellular signal-regulated kinase; GPx—glutathione peroxidase; GSE—grape seed extract; GSPE—grape seed proanthocyanidin extract; GSSG-R—glutathione reductase; MAPK—MAP kinase-activated protein kinase; NAFLD—non-alcoholic fatty liver disease; Nrf2—nuclear factor erythroid 2-related factor 2; PPARγ—peroxisome proliferator-activated receptor gamma; ROS—reactive oxygen species; SOD—superoxide dismutase; TNF-α—tumor necrosis factor alpha; TrxR—thioredoxin reductase. Compilation of data: [40,47,52,56,60,62,64,76].

## Data Availability

Data sharing not applicable.

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
