# Peer review of "A Review on Berry Seeds—A Special Emphasis on Their Chemical Content and Health-Promoting Properties"

_nutrients, 2023, doi:10.3390/nu15061422_

Round 1

Reviewer 1 Report

Journal- Nutrient

Manuscript ID: nutrients-2279190

Title-" A review on berry seeds a special emphasis on their chemical content and health-promoting properties"

1.     As per the binomial nomenclature, in any scientific documentation, first mention the full name of the organism (e.g., Rubus idaeus). Later, it can be written in short e.g., R. idaeus. Authors need to follow the rule of this classification system and maintain uniformity throughout the manuscript.

2.     In the section “Chemical content of berries and berry seeds,” the authors need to give the chemical structure of some common phytochemicals that possess high medicinal and nutritional values.

3.     Berry fruits are essential ingredients in the diet, herbal medicine, and dietary supplements. Like fruits, berry seeds contain a wide range of phytochemicals which may be used as ingredients in various preparations. These phytochemicals act as agonists or antagonists with various nuclear receptors such as PXR, CAR and AhR and regulate their downstream genes, such as CYP3A4, CYP2C9 and P-gp transporters and may induce herb-drug interaction. Authors can add information on “Herb-drug Interaction (HDIs)” and the risk of mediated berry seed phytochemicals. Authors can use the following articles as references DOI: https://doi.org/10.18433/J3NG6Z, https://doi.org/10.1016/j.jep.2022.115822; https://doi.org/10.1016/j.heliyon.2020.e05357; and DOI: 10.1007/s12272-011-1106-z.

Author Response

Thank you for reviewing the manuscript and for providing such helpful comments. All of them have been taken into consideration when revising the manuscript.

Title-" A review on berry seeds a special emphasis on their chemical content and health-promoting properties"

  1. As per the binomial nomenclature, in any scientific documentation, first mention the full name of the organism (e.g., Rubus idaeus). Later, it can be written in short e.g., idaeus. Authors need to follow the rule of this classification system and maintain uniformity throughout the manuscript.

Response: We have corrected the names.

  1. In the section “Chemical content of berries and berry seeds,” the authors need to give the chemical structure of some common phytochemicals that possess high medicinal and nutritional values.

Response: We have added chemical structures of selected compounds (Figure 2).

  1. Berry fruits are essential ingredients in the diet, herbal medicine, and dietary supplements. Like fruits, berry seeds contain a wide range of phytochemicals which may be used as ingredients in various preparations. These phytochemicals act as agonists or antagonists with various nuclear receptors such as PXR, CAR and AhR and regulate their downstream genes, such as CYP3A4, CYP2C9 and P-gp transporters and may induce herb-drug interaction. Authors can add information on “Herb-drug Interaction (HDIs)” and the risk of mediated berry seed phytochemicals. Authors can use the following articles as references DOI: https://doi.org/10.18433/J3NG6Z, https://doi.org/10.1016/j.jep.2022.115822; https://doi.org/10.1016/j.heliyon.2020.e05357; and DOI: 10.1007/s12272-011-1106-z.

Response: We have added information about herb-drug interactions (section 4. ‘Herb-drug interaction (HDI)’.

Reviewer 2 Report

The review is very interesting. Here are some comments and suggestions

  1. Table 1 is not clear. It is a very general summary, it would be better to have a table where the information is collected according to the type of berry, as indicated in Table 2. From my point of view, table 1 should be modified to indicate composition by berry type, and leave figure 1 as it is.

  1. The bibliography in the text is not unified in the way of expressing it. There are citations in which only the author is indicated, and others with the year. See line 167 and 176. everything should be the same. 

  1. Figure 2 is not necessary. it is not the objective of the article to study the different products marketed.

  1. Health-promoting properties of berry seeds - in vitro and in vivo studies: in some types of berries there is a lack of in vivo human trials, it is important that they appear.

  1. The last paragraph of 4.5.2. does not pertain to this point but to all, it should be made clearer.

Author Response

Thank you for reviewing the manuscript and for providing such helpful comments. All of them have been taken into consideration when revising the manuscript.

The review is very interesting. Here are some comments and suggestions:

  1. Table 1 is not clear. It is a very general summary, it would be better to have a table where the information is collected according to the type of berry, as indicated in Table 2. From my point of view, table 1 should be modified to indicate composition by berry type, and leave figure 1 as it is.

Response: We have modified the table.

  1. The bibliography in the text is not unified in the way of expressing it. There are citations in which only the author is indicated, and others with the year. See line 167 and 176. everything should be the same. 

Response: We have unified the bibliography.

  1. Figure 2 is not necessary. it is not the objective of the article to study the different products marketed.

Response: We think that availability of different products from berry seeds on the market is an important aspect that determines if a product can be realistically introduced to daily diet.

  1. Health-promoting properties of berry seeds - in vitro and in vivo studies: in some types of berries there is a lack of in vivo human trials, it is important that they appear.

Response: In some cases (raspberry, strawberry, cranberry), there are no clinical trials that study the seeds of these berries (same studies used whole fruits or combination of different seed extracts, and therefore did not meet the inclusion criteria). We have added a clinical trial that studied the activity of sea buckthorn seed oil: “A controlled trial by Vashishtha et al. has showed that sea buckthorn oil has positive effects on the cardiovascular system of healthy and hypertensive/hypercholesterolemic humans. An oral dose of 0.75 ml for 30 days reduced systolic and diastolic blood pressure, serum total cholesterol, triglycerides, and LDL-cholesterol compared to baseline values [74].

5.The last paragraph of 4.5.2. does not pertain to this point but to all, it should be made clearer.

Response: We have added information that this paragraph pertains to entire section 4.

Reviewer 3 Report

The review by Natalia SÅ‚awiÅ„ska and colleagues is focused on the biological properties of berry seeds, in the context of various biological activities, including anti-cancer, anti-hyperlipidemic, anti-obesity, and others. 

It is intriguing and I have a few questions:

1. The compounds deriving from fruits or agrifood-by products can play a role as antioxidant/radical sinkers in cancer. The authors should discuss its point (see. 

  1. Compr. Rev. Food Sci. Food Saf. 201110, 221–247.
  2.  Biochem. Pharmacol. 201598, 371–380.
  3. Cancers 202214(22), 5517; https://doi.org/10.3390/cancers14225517)
  4. 2. In the conclusions, is not clear if are ongoing clinical trials based on their use or not. If yes,  authors need to address the clinical trial status for various diseases.
  5. 3. It is known that some agrifood-derived compounds can bind EGFR.  Have the authors an idea about the molecular mechanism involving EGFR at the basis of the potential of barry seeds?
  6. (Cancers 202214(22), 5517; https://doi.org/10.3390/cancers14225517)
  7.  

Author Response

The review by Natalia SÅ‚awiÅ„ska and colleagues is focused on the biological properties of berry seeds, in the context of various biological activities, including anti-cancer, anti-hyperlipidemic, anti-obesity, and others. 

Thank you for reviewing the manuscript and for providing such helpful comments. All of them have been taken into consideration when revising the manuscript.

It is intriguing and I have a few questions:

  1. The compounds deriving from fruits or agrifood-by products can play a role as antioxidant/radical sinkers in cancer. The authors should discuss its point (see. 

Compr. Rev. Food Sci. Food Saf. 201110, 221–247.

 Biochem. Pharmacol. 201598, 371–380.

Cancers 202214(22), 5517; https://doi.org/10.3390/cancers14225517)

Response: We have added a paragraph on this topic “Numerous plant extracts and phytochemicals are being investigated as potential agents that could help with the treatment and prevention of civilization diseases, like cancer, diabetes, or cardiovascular diseases. Increased consumption of phenolic compounds has been linked with decreased occurrence of diseases related with oxidative stress. As most phenolic compounds show some degree of antioxidant or pro-oxidant activity, plants are a good source of compounds with potential beneficial effects that could be used as therapeutic agents. For example, many phenolics have shown anti-cancer properties thanks to either antioxidant (chemoprevention through reduction of DNA damage), or prooxidant (induction of cancer cell death through generation of reactive oxygen species) activity (Brewer, 2011; León-González et al., 2015; Sorrentino et al., 2022).”

  1. In the conclusions, is not clear if are ongoing clinical trials based on their use or not. If yes,  authors need to address the clinical trial status for various diseases.

Response: We have added information about clinical trials in conclusions: "There are multiple clinical trials (both completed and ongoing) that study the effect of grape seed extract on cardiovascular diseases, metabolic syndrome, or neurological disorders, but there are very few studies evaluating the efficiency of cranberry, raspberry, strawberry, and sea buckthorn seeds in vivo. There is a need for more clinical trials that would check the effectiveness and safety of less-known seeds, which would allow for full utilization of their potential.”

  1. It is known that some agrifood-derived compounds can bind EGFR.  Have the authors an idea about the molecular mechanism involving EGFR at the basis of the potential of barry seeds?

(Cancers 202214(22), 5517; https://doi.org/10.3390/cancers14225517)

Response: We have added new paragraph about this topic “Grape seed proanthocyanidins also showed anticancer properties. They inhibited the growth of MCF-7 (breast cancer) cell line (IC50 = 0.10 μM). They induced cell cycle arrest at the G2/M phase and increased the number of apoptotic cells. Moreover, proanthocyanidins decreased the expression of epidermal growth factor receptor (EGFR), which is involved in cancer proliferation, migration, invasion, and angiogenesis (Kong et al., 2021). In another study by Sun et al. grape seed proanthocyanidins inhibited the invasion of SCC13 (head and neck cutaneous squamous cell carcinoma) cells induced by epithelial growth factor (EGF), a known ligand and stimulator of EGFR (Sun et al., 2011).”